# An Emissions Arbitrage Algorithm to Improve the Environmental Performance of Domestic PV-Battery Systems

**Susan Isaya Sun** [1,*], **Andrew Frederick Crossland** [2], **Andrew John Chipperfield** [1] **and Richard George Andrew Wills** [1]

1   Faculty of Engineering and Physical Sciences, University of Southampton, Southampton SO17 1BJ, UK; a.j.chipperfield@soton.ac.uk (A.J.C.); rgaw@soton.ac.uk (R.G.A.W.)
2   Advance Further Energy Ltd., Retford DN22 6UF, UK; a.f.crossland@gmail.com or andrew.crossland@infratec.co.nz
*   Correspondence: ss3g15@soton.ac.uk

**Abstract:** Domestic PV-battery systems are rarely operated in a way which specifically maximizes environmental benefit. Consequently the studies that seriously examine such systems often find that the greenhouse gas and pollutant emissions savings of rooftop PV, though still positive, are lessened by adding a domestic battery. This study shows thatby simulating a PV-battery system with a range of sizes that this need not be inevitable. A novel algorithm was designed specifically to perform 'emissions arbitrage': to charge the battery when the grid emissions intensity is low and to discharge when it is high. It was found that the $CO_2$ saved relative to the same system with PV only can more than pay back the $CO_2$ debt of manufacturing the battery. This is true as long as the UK moves away from the present-day situation where natural gas-fired generators are nearly always the marginal generator. This work underlines the importance of both the operating strategy and the interactions between the system and the entire grid, in order to maximize the environmental benefit achievable with domestic PV-battery systems.

**Keywords:** GHG; $CO_2$; emissions arbitrage; solar PV; battery; algorithm

## 1. Introduction

There is growing interest in home battery products such as the Tesla PowerWall [1], Moixa Maslow [2], SolarWatt MyReserve [3], BYD B-Box [4], and many others. These complement rooftop solar PV arrays by storing excess generated energy that is not consumed onsite immediately, so that it can be used when consumption exceeds onsite generation. Reasons for the uptake of home batteries include the reduction of both electricity bills and dependence on incumbent energy suppliers, the use as backup power, and out of concern for the environment [5].

While much literature exists on the use of grid-connected domestic PV-battery systems for increasing energy self-sufficiency and reducing electricity bills [6–13], literature that examines environmental impacts shows that such systems cannot always be assumed to be 'green'. Kabakian et al. (2015) [14] showed that a 1.8 kW PV system with lead-acid batteries in Lebanon had slightly more embodied lifetime greenhouse gas (GHG) emissions than the 1.8 kW PV alone, 92 g of $CO_2$-equivalent per kWh delivered compared to 89 g/kWh. Jones et al. (2017) [15] found similarly that PV with battery would save 15% of a non-domestic building's $CO_2$ emissions, but PV alone would save 17%. Uddin et al. (2017) [16] reasoned similarly for lithium-ion batteries in the UK, on the basis that there are environmental impacts from manufacturing them, and additionally energy losses when charging/discharging. The same was found by Fares and Webber (2017) for Texas [17].

In contrast, Faria et al. (2014) [18] showed that a second-life electric vehicle battery could reduce global warming, abiotic depletion, acidification and eutrophication factors by 2% when used in a peak-shaving application in France, and 4–5% in a load-shifting application (both without PV). This is because the French grid emissions intensities change throughout the day in such a way that electricity is imported from the grid when emissions are low and exported to the grid when they are high. This 'emissions arbitrage' effect was not accounted for in the other papers mentioned, which assumed a constant grid emissions intensity.

A flaw in the work of Faria et al. (2014) [18] is that they used average rather than marginal emissions intensities. If some grid-generated electricity is displaced by the injection of PV power, or indeed any other intervention, not all the generator types (nuclear, coal, biomass, etc.) would have their output reduced in the same proportion as their total generation. The reduction would occur mostly for the generator type with the highest running cost. This gives rise to the concept of marginal emissions factor ($MEF$, as opposed to average emissions factor, $AEF$). Studies have shown that using $AEF$ rather than $MEF$ can cause errors of up to 25% [19–22].

It should also be noted that the battery operating strategies studied by Faria et al. (2014) were not designed to achieve emissions arbitrage. As such, environmental impacts were negative when those operating strategies were applied to Portugal and Poland [18]. The literature is abundant with algorithms for peak reduction, cost minimization and self-sufficiency maximization [6–13]. There is good reason for this: All these objectives are quantifiable and desired by consumers, distribution network operators, or other relevant stakeholders. However, environmental benefit is also desirable, as evidenced by survey data on opinions of renewable energy tariffs [23] and home energy storage [5]. Furthermore, it is now thought essential, in a special report by the IPCC [24], to limit global warming to 1.5 °C above the pre-Industrial average, and so to reduce GHG emissions accordingly.

It is not appropriate to equate energy self-sufficiency with environmental benefit (a link which Sun et al. (2018) [25] showed to be spurious), nor to regard the home battery as an environmental burden to be traded against a benefit, financial or otherwise [14,16,17]. This study seeks to show that it is possible for home batteries to achieve some environmental benefit even above PV without batteries, by judicious design of an emissions arbitrage algorithm.

To quantify the environmental benefits, it is necessary to produce a time series $MEF(t)$ which can be projected into the future. A constant average $MEF$ is insufficient as there is then no scope for emissions arbitrage. Historic $MEF(t)$ is also insufficient, as the grid technology mix may change drastically over the 25+ years lifetime of a PV array [26]. Bearing these requirements in mind, existing approaches to finding $MEF$ are reviewed in Section 2.

The methodology of this study is described in Section 3, including the modelled system setup, emissions arbitrage algorithm, sources of input data and how they are processed to calculate the system's environmental benefit, in the present day and in 2030 and 2050. Results are presented in Section 4, and discussed in Section 5, along with further work. The conclusions are in Section 6.

This study takes a markedly different approach from most of the existing literature on domestic energy storage, which either incorrectly assumes its environmental benefit, or infers a misalignment of the environmental objective with financial and other objectives. A notable exception is the work of McKenna et al. [27,28], who do consider time-varying $MEF(t)$ and the possibility of greater environmental benefits in a future decarbonized grid. The operating algorithm evaluated in their 2013 paper [27], however, was not designed specifically for emissions arbitrage. Furthermore, a new evaluation is timely, as they had analyzed the UK grid in 2009 (finding the life cycle impacts of a lead-acid battery to be equivalent to increasing an average household's energy consumption by 21%), and lead-acid has been overtaken by lithium-ion as the dominant home battery technology. Their 2017 presentation [28], which speculates on the effects of decarbonizing the grid on individual PV-battery systems, is still in need of supporting evidence. This study aims to provide such evidence.

It is shown in this study that by actively designing a new algorithm, environmental benefit can be achieved, as long as the time-varying nature of $MEF(t)$ is accounted for. It is by no means easy, and may still require changes in government policy to align the objectives. Even so, the rise of energy storage and demand response, in smart grids or virtual power plants, is revolutionizing renewable power by making it *dispatchable*: In other words, not constrained by uncontrollable forces such as weather. It is therefore vital to consider time-varying $MEF(t)$ in all such systems where algorithms are designed to control their dispatch, beyond just domestic PV-battery systems.

## 2. Literature Review of *MEF*

The problem is to calculate the marginal emissions factor $MEF(t)$, or better yet, the marginal generator response MGR(t), over time and in possible future scenarios with generation technology mixes different to today. The existing literature is reviewed against this goal. The main difficulty is that the problem deals with a hypothetical situation: For a hypothetical increase or decrease in total demand, what would the power generators do differently from otherwise? As the hypothetical situation never truly exists, the solution (or its validation) lies in finding appropriate proxies.

Bettle et al. (2006) [19] derived a fixed merit order by analyzing historical load duration curves of electricity generation plants in the UK. That is, the plants that output closer to their maximum for more of the time were identified as baseload and are thus higher in the merit order, whereas plants that output their maximum for less of the time were identified as load-following, or for even less of the time, as peaker plants, and occur lower in the merit order. The demand in each moment is then met by the plants in merit order, and the lowest-merit plant needed to fulfil the demand is the marginal generator. In reality, the merit order is not fixed but depends dynamically on running costs, electricity price, physical constraints, etc. Only a single average figure across the year for marginal $CO_2$ intensity was given with confidence, so this method could not be used to derive time-varying $MEF(t)$.

Hawkes (2010) [20] performed a linear regression of the hour-to-hour change in total $CO_2$ emissions ($\Delta E$) against the hour-to-hour change in total demand ($\Delta D$) of the UK grid, to derive $MEF$ as the gradient $\Delta E / \Delta D$ (kg/kWh). In essence, the proxy used for hypothetical changes in demand are real changes in demand $\Delta D$ at different times across the year. McKenna et al. (2017) [21] applied this method to the Irish grid by binning the data by total demand $D$ and performing linear regressions separately within each bin. This allowed them to produce time series of marginal $CO_2$ intensity. Siler-Evans et al. (2012) [22] did similarly for various grids in the USA, producing time series of marginal $SO_2$ and $NO_x$ as well as $CO_2$ intensity. However, being based on historical data, there was no clear way to extend this method to possible future generation mix scenarios.

Olkkonen and Syri (2016) [29] used the energy dispatch model EnergyPLAN to simulate operation of the Nordic grid. They ran the model once for a real year, and again for a hypothetical year with demand constantly greater than in the real year to a total of 1 TWh, finding the $MEF$ from the difference in resultant $CO_2$ emissions between them. While EnergyPLAN has been extensively peer-reviewed, the $MEF(t)$ time series produced was not validated against other methods.

The work by Lane Clark & Peacock and EnAppSys (2014) [30] on the other hand does compare four methods of producing $MGR$. These are: Merit order by running cost (out of all the final physical notifications given before gate closure), identifying the load-following plant (the one that increases or decreases output most closely in line with demand at the time-step in question and up to three half-hour periods before and after), highest bid or offer in the Balancing Mechanism (BM, a fairer reflection of merit order at delivery time rather than at gate closure an hour before), and running the Department of Energy and Climate Change Demand Dispatch Model (DECC DDM—similar in principle to the method of Olkkonen and Syri (2016) [29]). There were discrepancies between all the methods when applied to UK historic data, owing in part to plants running 'out of merit'. For example, running overnight despite high running costs because the costs of startup/shutdown are even greater. While LCP and EnAppSys judged the BM method the best, only the DDM method can be used to project future marginal generator responses, as the others all rely on historic data [30].

## 3. Method

The modelling of the domestic PV-battery system and emissions arbitrage algorithm is described here. It requires as input the marginal emissions factor time series $MEF(t)$, which is calculated for the UK in 2017 using an adaptation of Hawkes' method [20], and also for 2030 and 2050 using National Grid's Future Energy Scenarios [26]. This is then used to calculate the potential $CO_2$ arbitrage impact of the domestic PV-battery system. All Matlab code used in this study is available as the Supplementary folder 'matlab_code'. It was developed in Matlab R2018a.

In the following work only short-run impacts of avoided emissions were considered, as opposed to long-run impacts such as avoided construction of new peaker plants due to an intervention reducing peak electricity consumption enough to render such construction unnecessary. The latter were treated exogenously.

### 3.1. System Setup and Inputs

The system studied here is a single household PV array and home battery (Figure 1). For input to the simulation, electrical consumption data were collected at 5-min resolution from 1 February 2012 to 31 January 2013 by E.ON UK plc., for a house in the Midlands, UK, with annual consumption 3845 kWh, close to the present-day UK average of 3800 kWh. PV generation data were collected from 2 December 2015 to 30 November 2016 from a 3.6 kW array on a 45°-inclined southeast-facing roof in Berkshire, UK. It was averaged from 2-s to 5-min resolution, to match the load data. More detailed system parameters are given in Appendix A.

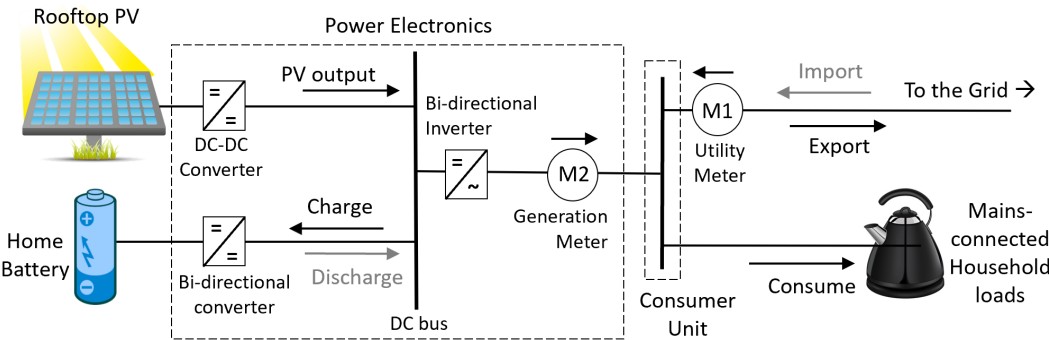

**Figure 1.** Schematic of DC-coupled domestic PV-battery system.

The base case PV capacity is set to 4 kW, which covers the annual consumption of 3845 kWh. Home batteries are available in a range of sizes, from the 2.2 kWh Moixa Maslow [2] to the 14 kWh Tesla PowerWall 2 [1], and so an intermediate value of 7.5 kWh was chosen for the base case. The effects of varying PV and battery capacities are shown in Section 4.2.

Generated power (MW) and total demand (MW) in the UK at 5-minute resolution for the year 2017 were downloaded from the GridWatch website [31]. This data source was chosen as being reliable, being synthesized from Elexon BM reports [32], but much more easily accessible (any desired time period from 2011 onwards can be downloaded as a CSV spreadsheet). The power generation data are aggregated by type: Coal, CCGT (combined cycle gas turbines), biomass, hydroelectric, wind, and various interconnectors. This was deemed a suitable level of granularity for application of Hawkes' method (Section 3.2).

To reduce the impact of missing data points, the GridWatch data were averaged from 5-min to half-hourly resolution. This left 97.5 h of data still missing out of a total 8760 h in 2017. Missing data were replaced by interpolation.

### 3.2. Adapted Hawkes' Method for $MEF(t)$

Although Hawkes' method [20] cannot be used to make future projections, some insight can be gained by applying it to the generation data of 2017. Siler–Evans' adaptation to Hawkes' method was followed in this work [22]:

Firstly, rather than linearly regressing the change in emissions $\Delta E$ from one time step to the next against the change in demand $\Delta D$, the change in power from each generator type was taken as the dependent variable. That is, $\Delta P_j$ was regressed against $\Delta D$, where $P$ is generated power (MW) from $j$ = coal, CCGT, biomass, nuclear, hydroelectric, French interconnector, Dutch interconnector (these being the most significant generator types, together supplying over 97% of the total demand [31]). Thus it is possible to apply conversion factors to obtain not just $CO_2$, but also $SO_2$, $NO_x$, and other marginal emissions impacts.

Secondly, in addition to binning the data by total demand $D(t)$, they were further binned by month. That is, instead of performing the linear regression on all the data in 2017, separate values of the gradient regressing $\Delta P_j$ on $\Delta D$ were obtained for each month and each demand bin ($D < 20$ GW, $20$ GW $\leq D < 22.5$ GW, ..., $D > 52.5$ GW). This way it was possible to capture seasonal effects, notably the decreased usage of coal in summer compared to winter. Note that to improve linear fit quality, run-of-river and pumped hydroelectric were summed to give the single category 'hydroelectric'. Only CCGT showed sufficient fit quality to perform the linear regression on single months; for the other generator types, the temporal bins are two-month periods.

From the gradients $\Delta P_j / \Delta D$, are derived time series of marginal generator response $MGR_j(t)$ (MW/MW). The time series of marginal emissions intensity is then given by:

$$MEF(t) = \sum_j MGR_j(t) \cdot c_j \qquad \text{(kg/kWh)} \qquad (1)$$

where $c_j$ is the intensity of emissions (for example $CO_2$ emissions, in kg/kWh) from generator type $j$.

Although $c_j$ can be substituted with the intensity of any type of emissions ($SO_2$, $NO_x$, even U235-equivalent for ionizing radiation), the principle is demonstrated here only with $CO_2$. The values used are those obtained by Staffell (2017) [33], listed in Table 1, and include the effects of typical load factors. They are for short-run emissions only, excluding embodied emissions of plant construction—those need to be added separately in a life cycle analysis, for example, of a PV-battery system. Although the energy absorbed by pumped hydroelectric is partly fossil-fuelled, this is included in the demand $D(t)$ and so the short-run $CO_2$ intensity of hydroelectric generated power is taken to be zero.

**Table 1.** Short-run $CO_2$-equivalent intensities of the most common generator types feeding into the UK grid [33].

| Generator Type | CO$_2$-eq Intensity (kg/kWh) |
| --- | --- |
| Coal | 0.937 |
| CCGT | 0.394 |
| Biomass | 0.120 |
| French interconnector | 0.053 |
| Dutch interconnector | 0.474 |
| Nuclear | 0.0 |
| Hydroelectric, wind, solar PV | 0.0 |

### 3.3. Emissions Arbitrage Algorithm

The emissions arbitrage operating algorithm designed for this study takes as input the $MEF(t)$ time series described in Section 3.2. Only generator dispatch up to 5 min previously is knowable, from which $MEF(t)$ is easily calculated from the previous month's dispatch data using the regression method of Section 3.2. The Supplementary Movie File 'algorithm_movie.mp4' explains the algorithm

(still frames in Appendix A). The battery is set to discharge at rate $\overline{P_B}$ until it is empty while the instantaneous $MEF(t)$ is above a given limit $L^+$, and to charge at $\overline{P_B}$ until it is full while instantaneous $MEF(t)$ is below limit $L^-$. When discharging, any excess power from both PV and battery above the load is exported (in Figure 2: 00:00–01:00, 05:40–10:55, 19:45–22:55), or if they are not enough, any remainder is imported to serve the load (01:00–05:40, 22:55–00:00). When charging, any excess PV power not used by the battery or load is exported (10:55–16:50 except 12:10–12:40 and 13:40–14:30), or if insufficient, power is imported from the grid to charge the battery (12:10–12:40, 13:40–14:30, 16:50–19:45). The limits $L^-$ and $L^+$ are continuously updated, set respectively at 0.02 kg/kWh below and above the mean of the previous 30 days' $MEF(t)$. In other words, the deadband is 0.04 kg/kWh.

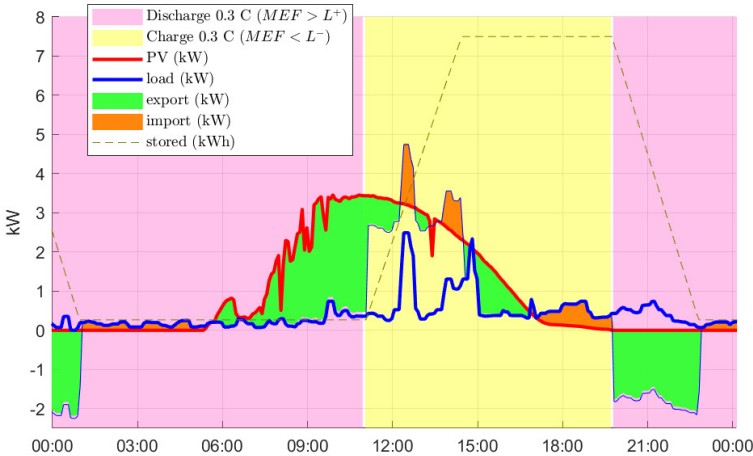

**Figure 2.** Operation of the emissions arbitrage algorithm for scenario CR50 (Section 3.4) with 4 kW PV, 7.5 kWh battery, $\overline{P_B} = 0.3$ C, 13 August.

The charge/discharge rate limit $\overline{P_B}$ is varied from 0.1–0.5 C (0.75–3.75 kW for a 7.5 kWh battery), and the effects shown in Section 4.2.

### 3.4. Future Grid

To derive $MEF(t)$ for future scenarios, the 2018 edition of Future Energy Scenarios (FES) [26] produced by National Grid (the UK's transmission system operator) was used. Four scenarios are presented, these having different characteristics in terms of degree of centralization and speed of decarbonization. This study limits itself to analyzing the two scenarios with fastest decarbonization, as only these allow the UK to meet its Paris Agreement target [34]: the more decentralized one is termed 'Community Renewables' (CR) and the more centralized one 'Two Degrees' (TD). Their demand characteristics and generator total capacities are given in Table 2, for 2030 and 2050, and also 2017 [35]. Note that the present-day UK grid consists of almost constant nuclear power as baseload, augmented by coal in winter. Despite significant wind and PV capacities, most UK electricity is still generated by CCGT, which performs most of the load-following, aided by a small amount of hydroelectric power. There is a move to replace coal with biomass, whose capacity remains small. Power can flow in the interconnectors to France, the Netherlands and Ireland, in either direction as determined by prices in each region [31].

**Table 2.** Demand characteristics and generator capacities in 2017 and the Community Renewables (CR) and Two Degrees (TD) scenarios of FES 2018, in 2030 (CR30, TD30) and 2050 (CR50, TD50) [35].

| | **2017** | **CR30** | **TD30** | **CR50** | **TD50** |
|---|---|---|---|---|---|
| Total demand (TWh) | 297 | 302 | 293 | 441 | 373 |
| Peak demand (GW) | 59 | 62 | 64 | 83 | 79 |
| Wind [a] (GW) | 17.6 | 47.0 | 39.4 | 83.2 | 65.7 |
| PV [b] (GW) | 12.4 | 33.0 | 24.3 | 66.2 | 43.7 |
| Biomass (GW) | 3.3 | 6.2 | 5.9 | 5.8 | 3.7 |
| Nuclear (GW) | 9.2 | 9.2 | 9.0 | 9.0 | 18.6 |
| CCS [c] (GW) | 0 | 0 | 0.9 | 0 | 12.1 |
| Hydroelectric, Marine and Other Renewables [d] (GW) | 3.6 | 7.3 | 8.2 | 11.2 | 13.9 |
| CCGT (GW) | 34.9 | 31.7 | 30.7 | 22.8 | 9.5 |
| Coal (GW) | 12.7 | 0 | 0 | 0 | 0 |
| Interconnector (GW) | 4 | 16.5 | 19.8 | 16.5 | 19.8 |
| V2G (GW) | 0 | 1.1 | 1.0 | 20.6 | 17.9 |
| Other Storage (GW) | 2.9 | 9.0 | 8.9 | 29.0 | 17.3 |

[a] Onshore and offshore wind are summed together. [b] As all PV is connected at distribution level, it is netted off from demand $D(t)$ rather than being its own category of generator in the dispatch model. [c] It is not specified in FES 2018 what combination of fossil fuels to biomass are to be combusted in carbon capture and storage (CCS) plants. The former emit some GHGs through leakage while the latter can have negative emissions under the right production and supply conditions [36]. The simplification is made here that CCS has zero short-run $CO_2$ emissions. [d] The 'Other Renewables' category in FES 2018 is here lumped together with hydroelectric and marine power. They are all assumed to have zero short-run $CO_2$ emissions, and the simplification is made that some combination of them can always be dispatched up to their total capacity.

A rule-based dispatch was used within these parameters to produce time series of power output by each generator type. The simplification of a fixed merit order was used for the future scenarios, going from low-carbon to higher-carbon generator types until the demand in that period $D(t)$ is satisfied. This is justified by an anticipated increase in the carbon price. Staffell (2017) [33] has shown the UK's carbon price floor to have already had an effect in terms of CCGT displacing coal in the merit order compared to as recently as 2012.

As total demand time series for each of the future scenarios was presently unavailable upon request, the demand for 2017, $D_{17}(t)$, is scaled and shifted such that the annual total and peak demand match those given in Table 2 for each scenario:

$$D_x(t) = a_x + b_x \cdot D_{17}(t) \tag{2}$$

for each scenario $x$ = CR30, TD30, CR50, TD50. Values for the constants $a_x$ and $b_x$ are given in Appendix B.

Wind and PV time series are taken from the 2017 Gridwatch data and scaled in proportion to their capacities in the future scenarios. In FES 2018 [26] it is assumed that power flows across interconnectors are only limited by total capacity, when in reality they depend on prices in the two interconnected regions. This simplification is kept in this study, with the further simplification that when exporting, all interconnectors have associated $CO_2$ intensity equal to that of France, and when importing, their $CO_2$ intensity is equivalent to that of the Netherlands. The reasoning is that some correlations in demand and renewable energy supply exist throughout Europe, so any export from the UK in future is likely to displace low-carbon generators (high in the merit order), and vice versa for imports to the UK. The interconnectors to France and the Netherlands transmit the greatest flows today compared to those to Ireland, and they have respectively low and high grid $CO_2$ intensity (Table 1). Although interconnections to Spain, Norway, Denmark, Germany, and Belgium are also anticipated in FES 2018 [26], the capacities of each are not revealed due to commercial sensitivity. Better accuracy than with the approximations used here is beyond the scope of this study.

Only nominal power (GW) of energy storage and vehicle-to-grid (V2G) were given in FES 2018 [26], but not their energy storage capacities (GWh). As such, there was not enough information to incorporate storage directly in the generator dispatch model in this study. However, storage is considered as an aggregation of home batteries as described in Section 3.5.

*3.5. Calculation for Domestic PV-Battery System*

This study calculates the annual $CO_2$ savings from the PV-battery system in 2017 and for CR and TD scenarios in 2030 and 2050. Two methods are employed to calculate annual $CO_2$ savings. Firstly, the generator dispatch model (Section 3.4) is run with no storage, and again with demand $D(t)$ greater by 1 MW constantly. The difference between them gives the marginal generator responses $MGR_j(t)$ from which $MEF(t)$ is calculated, similarly to the method of Olkkonen and Syri (2016) [29] and DECC DDM [30]. This $MEF(t)$ is fed into the PV-battery simulation along with a portion of $D(t)$ and PV generation in each scenario, as detailed in Table 3 (the loads, PV and batteries of the whole country are approximated as lumped together). The simulation output is used to adjust $D(t)$ to account for the presence of all the PV-battery systems. The generator dispatch model is then run again with the storage-adjusted demand as input. The difference in $CO_2$ emissions between the runs with and without storage is divided by the total PV capacity of participating PV-battery systems, and multiplied by 4 kW, to give the average contribution to $CO_2$ savings of a 4 kW PV system with batteries.

**Table 3.** Parameters involved in the demand adjustment due to energy storage in the future scenarios.

|  | CR30 | TD30 | CR50 | TD50 |
|---|---|---|---|---|
| Participating PV-battery systems (approx.) [a] | 2.40 mill. | 2.37 mill. | 7.73 mill. | 4.69 mill. |
| Participating PV capacity (GW) [b] | 9.60 | 9.49 | 30.93 | 18.77 |
| PV fraction [c] | 0.291 | 0.391 | 0.467 | 0.430 |
| Demand fraction [d] | 0.0256 | 0.0228 | 0.0698 | 0.0380 |

[a] Number of 7.5 kWh battery systems to bring the total up to the 'Other Storage' values in Table 2, assuming 2 h of storage capacity typical of commercially available home batteries today. For sensitivity testing of battery capacity, these values are decreased/increased inversely to battery capacity. [b] Assuming average PV capacity 4 kW peak for every 7.5 kWh of battery. For sensitivity testing of PV capacity, these values are increased/decreased in proportion to PV capacity. [c] Participating PV capacity in this table, divided by total PV capacity values in Table 2 (values depend on PV and battery capacities of single system as explained in points (a) and (b)). The remaining PV are not coupled with batteries. [d] Assuming profile of electrical loads participating in PV-battery systems is the same shape as total demand (that is, residential, commercial and industrial loads participate proportionally to their presence nationally), and that all systems are sized such that total PV-generated energy would equal total load throughout the year (values depend on PV and battery capacities of single system as explained in points (a) and (b)).

The second method takes the difference in emissions between the storage-adjusted demand described above, and that adjusted demand plus a further 1 MW constantly, again from running the generator dispatch model for each of those cases. The resultant $MEF(t)$ series is fed into the PV-battery simulation for a single PV-battery system with household electrical consumption as measured by E.ON UK plc. That is, a real single system is simulated as opposed to an aggregated system of all participating systems in the country. The grid export series thus obtained (import is counted as negative export) is multiplied element-wise by the $MEF(t)$ series interpolated to 5-min resolution. This gives the $CO_2$ savings of the single system in question, when operated simultaneously to many other such systems nationally. In other words, the first method gives the contribution of an average system on the national level while the second gives the marginal impact of the last, or *n*th, system installed.

**4. Results**

Presented here are graphs illustrating Siler–Evans' adaptation to Hawkes' method applied to 2017, results for future grid scenarios as battery C-rate limit $\overline{P_B}$ is varied, and as PV and battery capacity are varied.

### 4.1. Adapted Hawkes' Method on 2017

Figure 3 shows the goodness of fit $R^2$ for the linear regressions of $\Delta P_j$ against $\Delta D$ binned by month and demand $D(t)$, for each generator type $j$. $R^2 > 0.9$ is typical for CCGT, indicating a close correlation between changes in demand and changes in CCGT power in response. Exceptions occur in very low- and high-demand bins where there is little data. Especially in the summer months, $D(t) < 40$ GW typically, so there is no data at all in higher bins. A moderately good fit ($R^2 = 0.5$ to 0.8) is obtained for coal power response in winter. The poor correlation in summer is likely due to most coal plants outputting very low or zero power, so that they cannot ramp down further in response to decreasing $D(t)$. The goodness of fit is similar for hydroelectric power but throughout the year. Biomass and French and Dutch interconnectors responses are poorly correlated with changes in demand. They contribute little power (1–2 GW each to a demand ranging around 20–50 GW) and are unlikely to greatly distort the $CO_2$ $MEF(t)$ results. They are included for completeness.

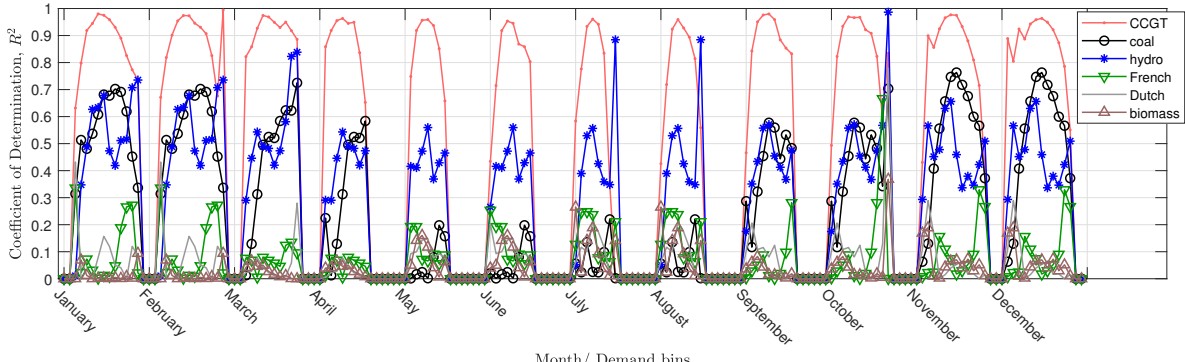

**Figure 3.** $R^2$ values for linear regressions of $\Delta P_j$ against $\Delta D$, for $j$ = CCGT, coal, hydroelectric, French and Dutch interconnectors, and biomass, binned by month and by demand $D$ (<20 GW, in 2.5 GW steps, to >52.5 GW). Nuclear is not included because it varies little, and as a result of repair/maintenance events or to fulfil an ancillary service contract, rather than in response to demand.

Samples of the $MGR(t)$ of the six generator types in summer and winter are shown in Figure 4. $MGR_{CCGT}(t)$ is high, cycling around 0.75 MW/MW, indicating most load-following is done by CCGT. The rest of the load-following is mostly contributed by coal and hydroelectric power, and intermittently by the French interconnector.

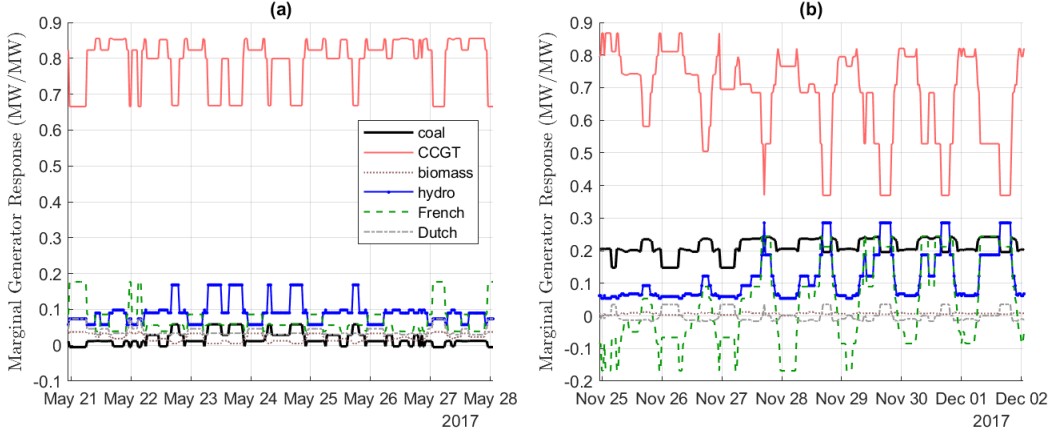

**Figure 4.** $MGR(t)$ of coal, CCGT, biomass, hydroelectric, and French and Dutch interconnectors, for (**a**) 21–28 May 2017, and (**b**) 25 November–2 December 2017.

Figure 5 shows for the same time periods as Figure 4 the results of applying $CO_2$ intensities for each generator type to the $MGR_j(t)$ series and summing them together. Variation can be between 0.35 and 0.5 kg/kWh in winter. Taking the average across the whole year of the $MEF$ in each half-hour period of the day results in the average day's $MEF(t)$, also shown in Figure 5. It varies <10% away from the mean 0.398 kg/kWh, because the patterns of low and high $MEF$ vary from day to day. This is consistent with the prediction by LCP/EnAppSys (2014) [30] that in the next few years the daily variation in $MEF(t)$ would reduce to close to zero, and appears to justify the use of a single value of $MEF$ in $CO_2$ abatement studies. As seen from Figure 5, however, the average day's $MEF(t)$ masks the variation that is present on other days.

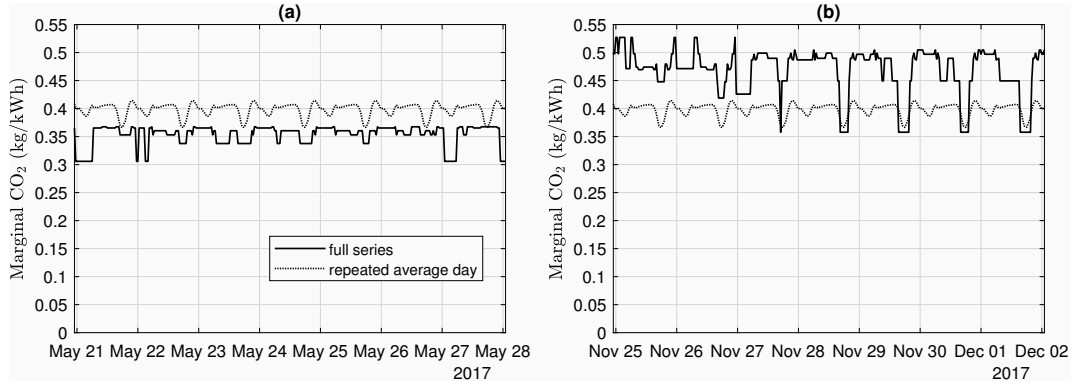

**Figure 5.** $MEF(t)$ and averaged across 2017 for each half-hour period of the day, for (**a**) 21–28 May 2017, and (**b**) 25 November–2 December 2017.

In fact, when the $CO_2$ savings are calculated for the base case 4 kW PV, 7.5 kWh battery system in 2017, there is a discrepancy between using the full $MEF(t)$ series and the average day's $MEF(t)$ repeated across the year. With the repeated average day, the annual $CO_2$ saving with 4 kW PV compared to without PV or battery is over-estimated at 1552 kg, whereas it is 1460 kg with the full $MEF(t)$ series. The repeated average day shows an annual disbenefit of 29.3 kg $CO_2$ with 7.5 kWh battery compared to with PV only, independent of C-rate limit. With the full $MEF(t)$ series, the battery benefit varies between −8.62 kg $CO_2$ (disbenefit) for 0.1 C, to +8.62 kg $CO_2$ (saved) for 0.4 C (Figure 6). This shows the importance of using the full $MEF(t)$ in calculations rather than studying only a single average day. The full $MEF(t)$ is used in all subsequent calculations.

### 4.2. Future Grid

The annual $CO_2$ saving was calculated for the base case 4 kW PV, 7.5 kWh battery system, in 2017 using Siler–Evans' adaptation to Hawkes' method, and in the future scenarios using the fixed merit-order generator dispatch model, as described in Section 3. Results for C-rate limit $\overline{P_B}$ varying between 0.1–0.5 C are shown in Figure 6.

The battery renders almost no benefit above PV only in 2017, whereas in the future scenarios, the battery benefit of the $n$th system is substantial even compared to the PV benefit (137 kg $CO_2$ saved annually in TD50, 0.1 C, up to 776 kg $CO_2$ in CR50, 0.5 C). The battery benefit of the $n$th system tends to increase with $\overline{P_B}$ (with the exception of 0.5 C in 2017). The opposite is true for the average system (note that this only compares the national system without and with home battery storage, so it is shown against the 'PV only' baseline). One possible explanation is that allowing charge/discharge at higher C-rate increases the system's emissions arbitrage capability. Especially when $MEF(t)$ varies rapidly, the battery may charge and discharge more fully when $\overline{P_B}$ is higher. This increases the $CO_2$ savings the $n$th system can achieve, but the increased utilization of all home batteries in the country at higher $\overline{P_B}$ causes more energy conversion losses, and increases the use of lower merit-order, more carbon-intense power generators during charging periods. Thus the average-system battery benefit decreases with $\overline{P_B}$, in some cases even negating the PV benefit.

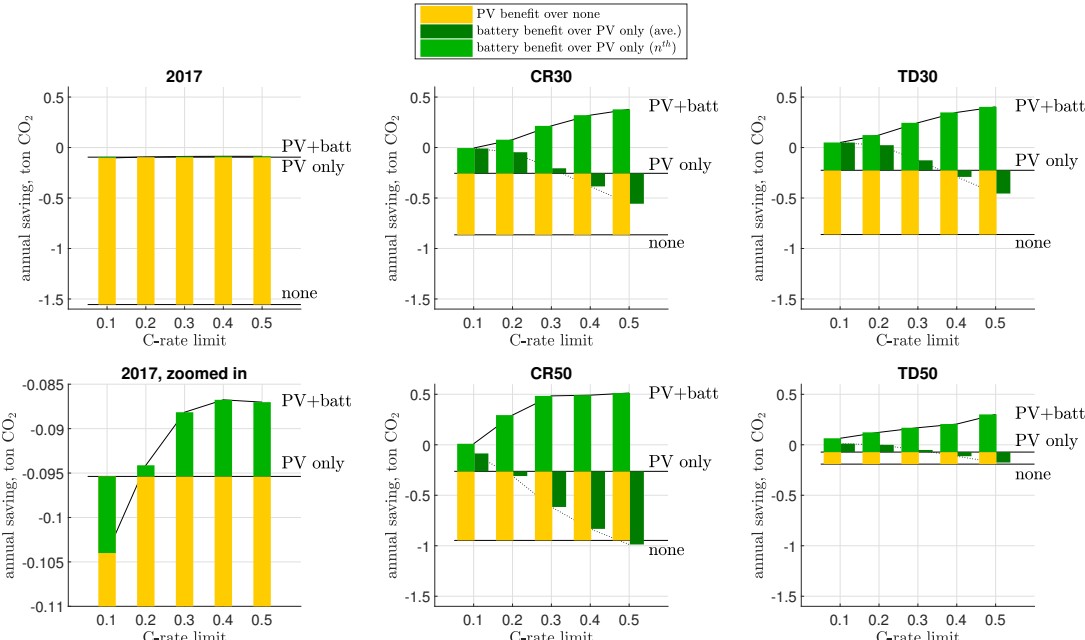

**Figure 6.** Annual $CO_2$ saving (tons) for varying C-rate limit, in 2017 and future scenarios, for average and $n$th systems. Savings are demarcated as benefit of PV only compared to no PV and no battery ('none'), and benefit of PV and battery above PV only. Zero $CO_2$ saved denotes carbon-neutrality for the household.

It is also possible, however, that the average-system results are erroneous because they do not take into account the interactions between individual PV-battery systems. Instead, the lumped system of all PV-battery systems in the country takes $MEF(t)$ assuming no storage nationally, as input to the emissions arbitrage algorithm. This is inaccurate, as in reality each individual system would take as input the $MEF(t)$ of the whole country, including all other PV-battery systems. To account for such behaviour is a task for further work.

The reason why such substantial $n$th-system battery benefit is achieved in the future scenarios is clear from examining the generators dispatched (Figure 7) and the resultant $MEF(t)$ (Figure 8). When there is so much renewable power on the grid, the marginal generator switches often between high-$CO_2$ CCGT or imports and low-$CO_2$ wind/biomass/other renewables. The result of this is greater variation in $MEF(t)$ (Figure 8 compared to Figure 5), which is exploited by the emissions arbitrage algorithm to achieve greater $CO_2$ savings. Note that these future scenarios, coupled with the assumption of generator dispatch in order from low- to high-carbon, represent cases which may not necessarily come to pass. A high carbon price floor may fail to be maintained; sufficient renewable generator capacity to meet the Paris Agreement target [34] may fail to be installed. The results would then be closer to those for 2017, or intermediate between them and the future scenarios studied.

Since there is a discrepancy between results for average and $n$th systems, the optimal C-rate limit $\overline{P_B}$ could not be determined conclusively. Therefore an intermediate value of $\overline{P_B} = 0.3$ C was taken for the PV/battery capacity sensitivity analysis. Results for 2017 and future scenarios are shown in Figure 9 ($n$th system only, as that is deemed more reliable than the average-system method).

In every case more $CO_2$ can be saved by installing more PV and more batteries. Although it is beyond the scope of this study to perform a full life cycle analysis, indicative estimates of $CO_2$ savings across the battery's lifetime can be obtained. In the 2030 cases in Figure 9, a household can save roughly 700 kg more $CO_2$ with 10 kWh of battery compared to without. This equates to 70 kg annually per kWh of battery, or 700 kg over a battery lifetime of 10 years [1,4], supposing such savings could be repeated yearly. Even in the TD50 case, the figure is more modest at 250 kg $CO_2$ saved per kWh over the 10-year battery lifetime. This is likely still sufficient to pay back the $CO_2$ debt of manufacturing a lithium-ion battery (various sources [37–39] estimate it to lie between 61–247 kg $CO_2$ per kWh).

If the grid mix of 2017 prevails, however, it will be almost impossible to pay back the battery's $CO_2$ debt, as annual $CO_2$ savings are almost independent of battery capacity in Figure 9, case 2017.

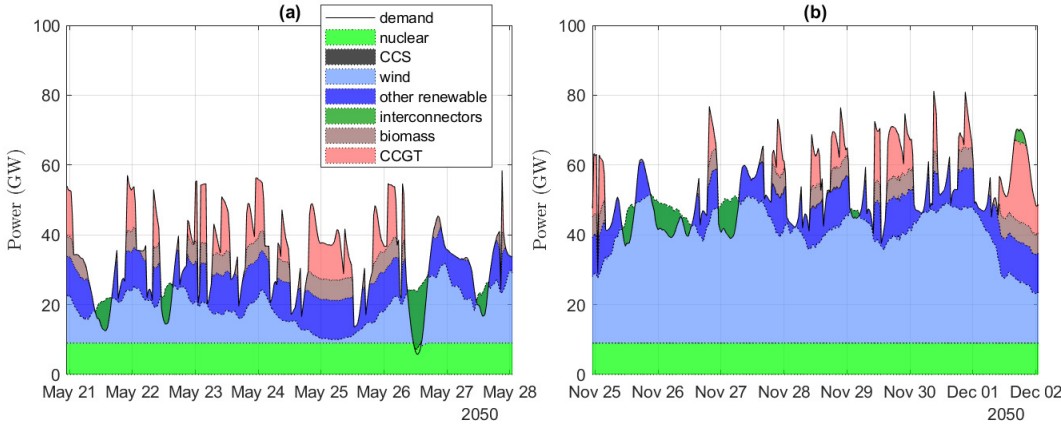

**Figure 7.** Generators dispatched in Community Renewables scenario, for (**a**) 21–28 May 2050, and (**b**) 25 November–2 December 2050. 4 kW PV, 7.5 kWh battery, C-rate limit 0.3 C.

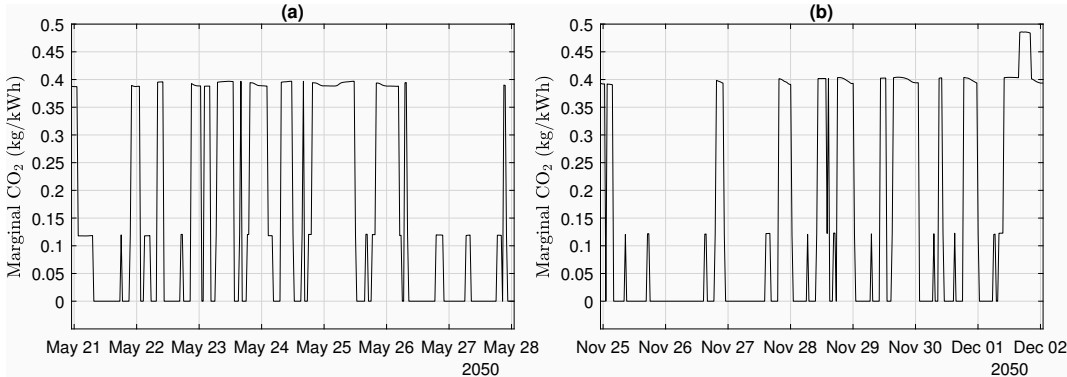

**Figure 8.** $MEF(t)$ in Community Renewables scenario, for (**a**) 21–28 May 2050, and (**b**) 25 November–2 December 2050. 4 kW PV, 7.5 kWh battery, C-rate limit 0.3 C.

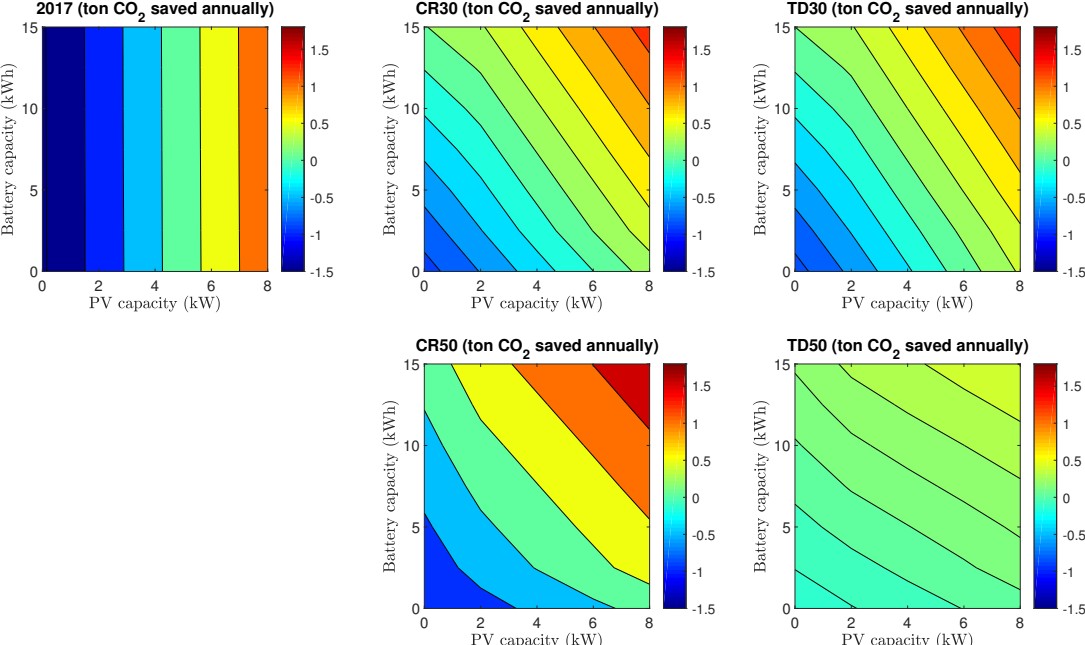

**Figure 9.** Annual $CO_2$ saving (tons) for varying PV and battery capacities, in 2017 and future scenarios (*n*th system only). Zero $CO_2$ saved denotes carbon-neutrality for the household.

## 5. Discussion

Using the adapted Hawkes' method to derive $MEF(t)$ for 2017, different results were found when using the full time series as opposed to the repeated average day's $MEF(t)$. This shows the importance of using the full time series, as the grid $CO_2$ variation is not adequately represented by an average day, nor is the average day representative of any single real day within the year.

It should be noted in Figure 6 that the $CO_2$ benefit of PV only is less in the 2030 scenarios than in 2017, and less again in 2050. That is, rooftop PV becomes less impactful as the grid decarbonizes. In contrast, the $n$th-system benefit of a home battery operating the emissions arbitrage algorithm is much greater in all the future scenarios than in 2017. This is the benefit of the battery above the same system with PV only. This challenges the previous literature [14–17,25,27] which finds home batteries to reduce the benefit of rooftop PV, showing the utmost importance of operating strategy design. If such emissions savings could be realized across the battery lifetime, the $CO_2$ debt of manufacturing it could be repaid and more. This is impossible, however, if the UK's grid technology mix continues as it is today, with the marginal generator response being almost wholly CCGT, as there is then insufficient variation in $MEF(t)$ for the algorithm to arbitrage effectively.

As system interactions were not accounted for, the results for the average system are not as reliable as those for the $n$th system. This would need to be addressed in future work. It could then be determined more conclusively whether a higher C-rate limit truly does yield greater $CO_2$ savings in the $n$th system but less when considering the average contribution of many PV-battery systems compared to the national system without any PV-battery. Doing so would also bridge the gap between short-run emissions impacts studied here, and long-run impacts on the national generating infrastructure [28]. If the environmental benefit still does not pay back the burdens of manufacturing and installing home batteries, other benefits must be seriously weighed against lifetime environmental impacts. These were not addressed in this study, but include avoidance of distribution losses in cases where these may be high for PV export to the grid [25], and reduction of voltage violations on the distribution network [40].

Although the battery benefit is greater in the lower-carbon future scenarios, it becomes small again in TD50, a scenario characterised by over twice as much nuclear generating capacity (18.6 GW) than the others, and 12.1 GW of CCS. $MEF(t)$ does not vary enough when there is so much low-carbon power on the grid. It must be hoped that improvements in battery lifetime and efficiency keep pace with progress in grid decarbonization. The emissions arbitrage algorithm itself may also be improved, perhaps using dynamic programming with forecasting of generation and demand, to optimally schedule the battery dispatch, rather than blindly charging/discharging until full/empty.

This study does not constitute a life cycle analysis. For this, it would be necessary to define the system boundaries and functional unit, for example, kWh delivered [41]. The environmental impacts across the whole system lifetime must be calculated, including manufacture, operation and disposal/recycling of all components, and then divided by the total energy supplied to the household loads and exported to the grid, to find the impacts per kWh. This must be done for other impact categories in addition to global warming. Even though a $CO_2$ benefit appears possible by operating the battery with the emissions arbitrage algorithm, the same may or may not be true for $SO_2$, $NO_x$ or human toxicity, for example. Further, it will be necessary to account for degradation of the battery and PV over the system lifetime.

The validity of the future scenarios $MEF(t)$ are only as good as the dispatch modelling used to obtain them. A highly simplified fixed merit order was assumed in this study: The advantage of transparency and easy reproducibility comes at the cost of no account being taken of price dynamics, generator ramping or part-loading, nor transmission constraints. Since National Grid ran the BID3 dispatch model for their FES 2018 analysis [26], which takes these factors into account, an obvious next step would be to apply the adapted Hawkes' method to their BID3 results, to see if $MEF(t)$ differs greatly from that obtained here. However, it has not been possible to obtain the BID3 results required.

Finally, it would be valuable to conduct financial analyses of the PV-battery system running the emissions arbitrage algorithm. Such information, together with visualizations like those in Figure 9, can help a customer choose the PV-battery capacities combination to maximize $CO_2$ savings within the constraints of their initial investment. This would depend on the customer's household load profile, and so it would be valuable to analyze the effects of above- and below-average consumption. A lower net present value and/or internal rate of return, or longer payback period compared to a PV-only system may be tolerated by some particularly environmentally conscious consumers. However, a change in policy may be necessary to align the environmental and financial incentives, for example, linking a subsidy to kg $CO_2$ (or other emissions) saved. With smart meters and home batteries themselves commonly recording data at 15-min resolution or better, the only issues would be a need to standardize the calculation of $MEF(t)$ and the verification of each household's emissions savings in a way that respects data privacy. Another possibility is for energy suppliers to offer a time-of-use tariff dependent on $MEF(t)$. Considering Wadebridge Renewable Energy Network's Sunshine tariff trial (https://wren.uk.com/sunshine) and Green Energy UK's Tide tariff (https://www.greenenergyuk.com/tide), this is already an idea with commercial applications.

## 6. Conclusions

In this study, a novel PV-battery operating algorithm was designed specifically to perform emissions arbitrage: Charging when grid emissions intensity is low and discharging when it is high. To accurately quantify its environmental performance, it was necessary to use:

- The marginal rather than average emissions intensity,
- A full year time series rather than a constant value, or an average day,
- A projection of the future power grid rather than only the present-day.

These goals were achieved by applying Siler–Evans' adaptation [22] to Hawkes' method [20] to UK 2017 data from Gridwatch [31] to find $MEF(t)$ for the present day. For the future, National Grid's FES 2018 [35] was used to parametrize a fixed merit order generator dispatch model.

It is shown to be possible for a home battery to fully repay its $CO_2$ debt of manufacture by operating the emissions arbitrage algorithm. In contrast to previous literature which finds an environmental benefit associated with domestic PV-battery, but less than that of PV alone [14–17,27], the benefit found here is for the system with battery relative to without. However, this benefit is contingent on $MEF(t)$ varying sufficiently, underlining the importance of decarbonizing the whole electricity grid. Otherwise this variance will not be achieved if CCGT remains on the margin nearly all the time.

Further work was identified:

- Use an improved dispatch model for future scenarios,
- Correctly account for interactions between PV-battery systems,
- Improve the algorithm,
- Conduct a life cycle analysis, including environmental impacts other than only GHG emissions,
- Conduct a financial analysis, with a range of household load profiles, to identify if policy changes are required to align financial and environmental objectives.

In closing, a rooftop PV with home batteries is one example of a dispatchable decarbonizing intervention; other researchers are encouraged to apply the methods presented here to design emissions arbitrage algorithms and evaluate their performance for other systems: Commercial- or utility-scale PV or wind farms, or other renewable power sources, in combination with batteries, compressed air, pumped hydroelectric, or other forms of energy storage, or demand response. To aid this endeavour, all Matlab code and a spreadsheet of numerical results are available as Supplementary Files.

**Supplementary Materials:** The following are available online at http://www.mdpi.com/1996-1073/12/3/560/s1, Matlab code folder: matlab_code, Spreadsheet: GridCarbonResults.xls, Video: algorithm_movie.mp4.

**Author Contributions:** S.I.S. conceived of the idea to use the time-varying nature of grid emissions intensity to improve the environmental impacts of PV-battery operation. She developed the code, sourced input data, analyzed results, and wrote the original draft, as part of a PhD project under the supervision and funding of R.G.A.W. and A.J.C. The emissions arbitrage algorithm is based on an idea by A.F.C., developed and implemented in code by S.I.S. The review and editing of the manuscript were carried out by A.F.C. and R.G.A.W.

**Funding:** This work was supported by the RCUK's Energy Programme as part of the research project 'Joint UK-India Clean Energy Centre (JUICE)' [grant ref: EP/P003605/1]; and the EPSRC Centre for Doctoral Training (CDT) in Energy Storage and its Applications [grant ref: EP/L016818/1].

**Acknowledgments:** PV generation data kindly supplied by Dickon Hood; household demand data by E.ON UK plc. With special thanks to John Barton, Ian Cole, and Oytun Babacan, for helpful conversations.

**Conflicts of Interest:** The funders had no role in the design of the study; in the collection, analyses, or interpretation of data; in the writing of the manuscript, or in the decision to publish the results.

## Abbreviations

The following abbreviations are used in this manuscript:

| | |
|---|---|
| 17 | Year 2017 (as subscript, $x = 17$). |
| $AEF$, $AEF(t)$ | Average Emissions Factor (value or time series), kg/kWh. |
| bio | Biomass (as subscript, $j = $ bio). |
| BM | Balancing Mechanism. |
| CCGT | Combined Cycle Gas Turbines (as subscript, $j = $ CCGT). |
| CCS | Carbon Capture and Storage (as subscript, $j = $ CCS). |
| CR30, CR50 | Community Renewables demand scenario, in 2030, 2050 (as subscript, $x = $ CR30). |
| $D_x(t)$ | Demand time series in scenario $x$, MW. |
| DECC | Department of Energy and Climate Change. |
| DDM | Demand Dispatch Model. |
| FES | Future Energy Scenarios. |
| Fr | French interconnector (as subscript, $j = $ Fr). |
| GHG | Greenhouse Gas, kg $CO_2$-equivalent. |
| IPCC | Intergovernmental Panel on Climate Change. |
| $j$ | (As subscript) generator type. |
| $L^+$, $L^-$ | Upper, lower, limits defining operating mode of emissions arbitrage algorithm, kg/kWh. |
| $MEF$, $MEF(t)$ | Marginal Emissions Factor (averaged or time-varying instantaneous), kg/kWh. |
| $MGR_j(t)$ | Marginal Generator Response for generator type $j$, MW/MW. |
| $\overline{P_B}$ | Battery charge/discharge limit in emissions arbitrage algorithm, C-rate. |
| $P_j(t)$ | Power from generator type $j$, MW. |
| PV | Photovoltaic. |
| TD30, TD50 | Two Degrees demand scenario, in 2030, 2050 (as subscript, $x = $ TD30). |
| $x$ | (As subscript) demand scenario. |

## Appendix A. PV-Battery System Details

The home battery simulated in this study was roughly modelled on the Tesla PowerWall. The parameters given in Table A1 could not be found directly from the technical specifications [1], which only state a round-trip efficiency of 90%, and maximum power rating 7 kW for a 14 kWh PowerWall 2, that is, 0.5 C. The efficiencies of the battery and power electronics were set at constant levels typical of what can be expected in the industry. They do vary, however, depending on the power throughput. This complexity is not modelled here. The state of charge (SoC) limits are such because most lithium-ion battery products set voltage cut-off limits to preserve battery life. In this case we are guided by 13.5 kWh capacity available out of a nominal 14 kWh for a Tesla PowerWall 2. The battery self-discharge includes both loss of charge of the battery itself (occurring over several months) and power for the battery management system. A rate of 0.001 C is equivalent to 7.5 W for a 7.5 kWh battery, which is considered reasonable for the control electronics involved.

The parameters in Table A1 will change over time as innovations occur. They can be updated in the PV-battery simulation by editing the function 'init.m', which is included in the Supplementary folder 'matlab_code'.

**Table A1.** Parameters used for the battery in the PV-battery simulation.

| Parameter | Value |
|---|---|
| Battery converter 1-way efficiency (%) | 97 |
| Battery inverter 1-way efficiency (%) | 97 |
| Battery charge/discharge 1-way efficiency (%) | 98.5 |
| Battery upper SoC limit (%) | 100 |
| Battery lower SoC limit (%) | 3.57 |
| Battery self-discharge (C-rate) | 0.001 |

The Supplementary movie file 'algorithm_movie.mp4' explains the operation of the emissions arbitrage algorithm. Figures A1, A2, and Figure 2 in Section 3.3 show still frames from the movie.

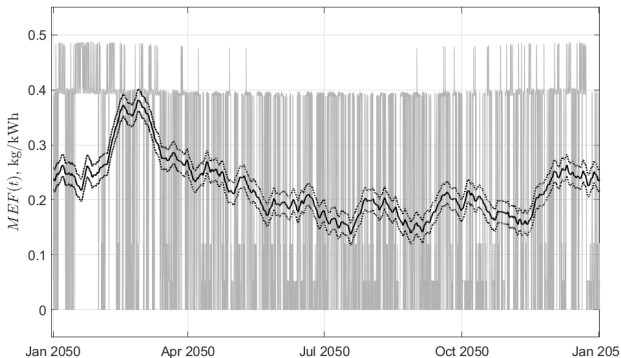

**Figure A1.** $MEF(t)$ across the year in scenario CR50, shown with its moving mean of the previous 30 days, and limits $L^+$ and $L^-$ which are respectively 0.02 kg/kWh above/below the moving mean.

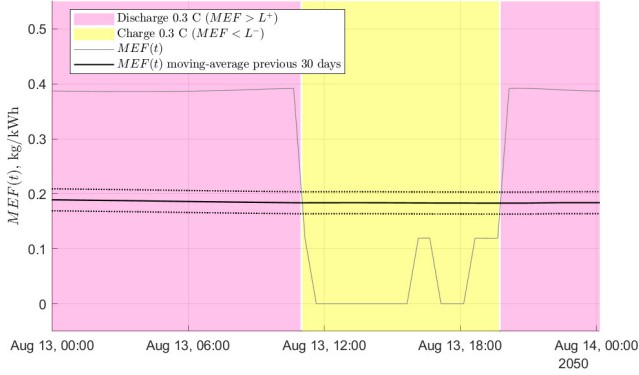

**Figure A2.** The above figure zoomed in to 13 August, showing charge and discharge modes of the battery, defined respectively as when $MEF(t) < L^-$ and $MEF(t) > L^+$.

See Figure 2 in Section 3.3 for how the charge/discharge modes thus defined in the emissions arbitrage algorithm interact with the real-time PV generation and household electricity consumption. The result is the import/export of electricity from/to the grid as required.

**Appendix B. Scaling and Shifting $D(t)$ for Future Scenarios**

In Section 3.4, the demand for 2017, $D_{17}(t)$, is scaled and shifted such that the annual total and peak demand match those given in Table 2 for each scenario:

$$D_x(t) = a_x + b_x \cdot D_{17}(t) \tag{A1}$$

for each scenario $x = $ CR30, TD30, CR50, TD50. Parameters $a_x$ and $b_x$ are obtained by solving simultaneous equations:

$$\text{Total demand (GWh)} = 8760\,\text{h} \times a_x + 297{,}000\,\text{GWh} \times b_x \tag{A2}$$

$$\text{Peak demand (GW)} = a_x + 59\,\text{GW} \times b_x \tag{A3}$$

Their solutions are given in Table A2.

**Table A2.** Scaling and shifting parameters $a_x$ and $b_x$ to transform 2017 demand series $D_{17}(t)$ into series appropriate for future scenarios in terms of total and peak demand.

|  | 2017 | CR30 | TD30 | CR50 | TD50 |
|---|---|---|---|---|---|
| Total demand (TWh) | 297 | 302 | 293 | 441 | 373 |
| Peak demand (GW) | 59 | 62 | 64 | 83 | 79 |
| $a_x$ (GW) | 0 | −2.71 | −7.83 | +6.22 | −6.62 |
| $b_x$ | 1 | 1.10 | 1.22 | 1.30 | 1.45 |

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
