# Peer review of "An Emissions Arbitrage Algorithm to Improve the Environmental Performance of Domestic PV-Battery Systems"

_energies, doi:10.3390/en12030560_

Round 1

Reviewer 1 Report

Well-written, timely, good knowledge of literature and clearly advances state of the art, and of high relevance to academia, policy, and industry. Good pointers to where future research should focus. A good paper that demonstrates what research in this area should aim to achieve. A few minor comments:

How would the results be affected by changes to the demand profile of the household? If they were a naturally high or low self-consumer, would this change the impact of the battery?

The dispatch model used for future scenarios is idealised, assuming that the merit-order is equal to the carbon order. This is the best case scenario for maximising the variability of MEF, and so produces best possible results for the algorithm. The authors should state that any deviation of the merit order from this ideal will reduce the effectiveness of the algorithm. The results are not conservative estimates therefore.

Figure 8 – shows renewables operating as the marginal generator for substantial periods of time, and is the reason why the algorithm produces such favourable environmental benefits – because the battery can charge with renewables as the marginal generator, and discharge to displace fossil fuel as the marginal generator later. Currently non-dispatchable renewables do not operate on the margin (unless they are being dispatched down or curtailed). Is it feasible to assume non-dispatchable renewables such as wind can operate on the margin and act as the generation technology that balances demand? I think you need to provide an argument to support this assumption.

Author Response

To the Reviewers, to Ms. Lv, Ms. Wu, and all involved in the review and editing of ‘An Emissions Arbitrage Algorithm to Improve the Environmental Performance of Domestic PV-Battery Systems’:

On behalf of all the authors, thank you kindly for some very encouraging and insightful feedback. We address the individual comments of each Reviewer in turn, below. We hope you will find our proposed changes satisfactory; we certainly believe the paper is improved by making these changes in response to the Reviewers’ comments.

Yours sincerely,

S. I. Sun

Responses to Reviewer 1

Well-written, timely, good knowledge of literature and clearly advances state of the art, and of high relevance to academia, policy, and industry. Good pointers to where future research should focus. A good paper that demonstrates what research in this area should aim to achieve. A few minor comments:

Point 1. How would the results be affected by changes to the demand profile of the household? If they were a naturally high or low self-consumer, would this change the impact of the battery?

Response 1: This is an excellent point and will be addressed in a future publication. We propose inserting in line 400 of the discussion: “This would depend on the customer's household load profile, and so it would be valuable to analyze the effects of above- and below-average consumption.”, and into line 432 of the conclusions: “Conduct a financial analysis, with a range of household load profiles,…”.

Point 2. The dispatch model used for future scenarios is idealised, assuming that the merit-order is equal to the carbon order. This is the best case scenario for maximising the variability of MEF, and so produces best possible results for the algorithm. The authors should state that any deviation of the merit order from this ideal will reduce the effectiveness of the algorithm. The results are not conservative estimates therefore.

Point 3. Figure 8 – shows renewables operating as the marginal generator for substantial periods of time, and is the reason why the algorithm produces such favourable environmental benefits – because the battery can charge with renewables as the marginal generator, and discharge to displace fossil fuel as the marginal generator later. Currently non-dispatchable renewables do not operate on the margin (unless they are being dispatched down or curtailed). Is it feasible to assume non-dispatchable renewables such as wind can operate on the margin and act as the generation technology that balances demand? I think you need to provide an argument to support this assumption.

Responses 2, 3: We propose to address the above two points by inserting after line 333: “Note that these future scenarios, coupled with the assumption of generator dispatch in order from low- to high-carbon, represent cases which may not necessarily come to pass. A high carbon price floor may fail to be maintained; sufficient renewable generator capacity to meet the Paris Agreement target [34] may fail to be installed. The results would then be closer to those for 2017, or intermediate between 2017 and the future scenarios studied.

(Much greater curtailment of renewables can be expected if their capacity becomes as great as in Table 2, but the Reviewer is correct, we needed to clarify that we did not assign any particular likelihood to this happening.)

We thank the Reviewer for the very helpful and encouraging feedback.

Responses to Reviewer 2

This article is very well developed, both from scientific and presentation point of view. At the same time, it presents very interesting results, useful for researchers and users of PV systems.

I recommend the publishing of the paper as it is.

Response: We thank the Reviewer for their supportive feedback.

Responses to Reviewer 3

This is a very thorough and interesting study of the benefits of emissions arbitrage for PV-battery systems attached to the UK grid. The paper shows that provided the grid emissions change with time sufficiently, then arbitrage can give useful savings in COemissions. This is not the case at present, but will be in the future. This is a well-presented paper that should be published. The supplementary video clearly summarises how the algorithm works. I have only a few suggestions for the authors to consider:

Point 1. Change lines 10 and 11 to - This is not the case when the penetration of renewables on the grid is high and underlines the importance of the operating strategy, in order to maximise the ….

[This is to avoid the uncertainty as to how far the grid is decarbonized, as there is clearly no arbitrage when all grids are fully decarbonized]

Response 1: This is a well-spotted and very subtle point. It is true that emissions arbitrage would not work if a grid was already fully decarbonized – but what if the grid was brought to this point by installing such battery systems? We hope the Reviewer will permit a compromise, “This work underlines the importance of both the operating strategy and the interactions between the system and the entire grid, in order to maximize…” on lines 10-11.

Point 2. Line 23- always considered ‘green’ i.e without emissions. But changing from 89 to 92 g/kWh is a very small addition that on its own might be thought green?

Response 2: These figures are for lifetime emissions rather than saved emissions. We propose amending lines 24-26 to clarify this: “Kabakian et al. (2015) [14] showed that a 1.8 kW PV system with lead-acid batteries in Lebanon had slightly more embodied lifetime greenhouse gas (GHG) emissions than the 1.8 kW PV alone, 92 g of CO_2-equivalent per kWh delivered compared to 89 g/kWh”.

Point 3. Line 53- It is now thought essential, in a new report by the IPCC, to limit global warming to 1.5 oC

Response 3: Change accepted. A reference to this IPCC Special Report and a glossary entry for ‘IPCC’ will be inserted.

Point 4. Discussion starting on line 307. I would have expected maximising environmental benefit would be well correlated with cost benefit, particularly when the penetration of renewables on the grid is high, as the marginal cost of wind and solar is very low - as are their emissions.

Response 4: Intuitively it might be expected that integrating wind and solar on the grid is a means to correlate environmental and economic benefits. However this is, justifiably, yet to be widely accepted in academia and industry. Although wind and solar costs are very low compared to historic levels, there is much misinformation in the market and care should be taken to assess cost estimates. For example, solar is not far from achieving grid parity in the UK, meaning some projects will be profitable whilst others will not. The same is true of wind. Similarly, the commercial benefits of solar/wind for consumers are not just related to the cost of generation, but also the cost of other technologies e.g. storage, the shaping of demand, and the influence this has on marginal plant, etc. We do not wish to go into that detail in the paper, and it is important to recognize that although the cost of generation may vary, many people in the UK remain on fixed-price tariffs which do not directly reflect underlying changes in wholesale electricity generation prices. In such a case, no price arbitrage would be happening (simultaneously to emissions arbitrage or otherwise) and in fact the energy losses from charging/discharging the battery would cause the bill to be higher.

Point 5. Line 377- the life cycle emissions of batteries will decrease as the world’s energy systems are decarbonized.

What is the role of the battery systems in storing excess generation by wind and solar, particularly when their penetration on the grid is high?

Response 5: Storing excess solar generation when grid penetration is high is what the emissions arbitrage algorithm is designed to do. Although we focused on a domestic context, the concept should be applicable to commercial/industrial buildings and utility-scale renewable generator parks, as stated in lines 436-438 of the conclusions.

We thank the Reviewer for the helpful and thought-provoking feedback.

Reviewer 2 Report

This article is very well developed, both from scientific and presentation point of view. At the same time, it presents very interesting results, useful for researchers and users of PV systems.

I recommend the publishing of the paper as it is.

Author Response

(The authors gave the same response as above.)

Reviewer 3 Report

This is a very thorough and interesting study of the benefits of emissions arbitrage for PV-battery systems attached to the UK grid. The paper shows that provided the grid emissions change with time sufficiently, then arbitrage can give useful savings in CO2 emissions. This is not the case at present, but will be in the future. This is a well-presented paper that should be published. The supplementary video clearly summarises how the algorithm works. I have only a few suggestions for the authors to consider:

Change lines 10 and 11 to - This is not the case when the penetration of renewables on the grid is high and underlines the importance of the operating strategy, in order to maximise the ….

[This is to avoid the uncertainty as to how far the grid is decarbonized, as there is clearly no arbitrage when all grids are fully decarbonized]

Line 23- always considered ‘green’ i.e without emissions. But changing from 89 to 92 g/kWh is a very small addition that on its own might be thought green?

Line 53- It is now thought essential, in a new report by the IPCC, to limit global warming to 1.5 oC

Discussion starting on line 307. I would have expected maximising environmental benefit would be well correlated with cost benefit, particularly when the penetration of renewables on the grid is high, as the marginal cost of wind and solar is very low - as are their emissions.

Line 377- the life cycle emissions of batteries will decrease as the world’s energy systems are decarbonized.

What is the role of the battery systems in storing excess generation by wind and solar, particularly when their penetration on the grid is high?

Author Response

(The authors gave the same response as above.)
